# Extraction of Tropical Fruit Peels and Development of HPMC Film Containing the Extracts as an Active Antibacterial Packaging Material

**DOI:** 10.3390/molecules26082265

**Published:** 2021-04-14

**Authors:** Tanpong Chaiwarit, Nutthapong Kantrong, Sarana Rose Sommano, Pornchai Rachtanapun, Taepin Junmahasathien, Mont Kumpugdee-Vollrath, Pensak Jantrawut

**Affiliations:** 1Department of Pharmaceutical Sciences, Faculty of Pharmacy, Chiang Mai University, Chiang Mai 50200, Thailand; tanpong.c@cmu.ac.th (T.C.); phoenixj035@hotmail.com (T.J.); 2Department of Restorative Dentistry, Faculty of Dentistry, Khon Kaen University, Khon Kaen 40002, Thailand; natthkan@kku.ac.th; 3Research Group of Chronic Inflammatory Oral Diseases and Systemic Diseases Associated with Oral Health, Faculty of Dentistry, Khon Kaen University, Khon Kaen 40002, Thailand; 4Plant Bioactive Compound Laboratory (BAC), Department of Plant and Soil Sciences, Faculty of Agriculture, Chiang Mai University, Chiang Mai 50200, Thailand; sarana.s@cmu.ac.th; 5Cluster of Research and Development of Pharmaceutical and Natural Products Innovation for Human or Animal, Chiang Mai University, Chiang Mai 50200, Thailand; 6Division of Packaging Technology, School of Agro-Industry, Faculty of Agro-Industry, Chiang Mai University, Chiang Mai 50100, Thailand; pornchai.r@cmu.ac.th; 7Cluster of Agro Bio-Circular-Green Industry (Agro BCG), Chiang Mai University, Chiang Mai 50100, Thailand; 8Department of Pharmaceutical Engineering, Beuth University of Applied Sciences Berlin, 13353 Berlin, Germany; Vollrath@beuth-hochschule.de

**Keywords:** mangosteen, rambutan, mango, tropical fruit extract, antibacterial, film

## Abstract

In recent years, instead of the use of chemical substances, alternative substances, especially plant extracts, have been characterized for an active packaging of antibacterial elements. In this study, the peels of mangosteen (*Garcinia mangostana*), rambutan (*Nephelium lappaceum*), and mango (*Mangifera indica*) were extracted to obtain bioactive compound by microwave-assisted extraction (MAE) and maceration with water, ethanol 95% and water–ethanol (40:60%). All extracts contained phenolics and flavonoids. However, mangosteen peel extracted by MAE and maceration with water/ethanol (MT-MAE-W/E and MT-Ma-W/E, respectively) contained higher phenolic and flavonoid contents, and exhibited greater antibacterial activity against *Staphylococcus aureus* and *Escherichia coli*. Thus, both extracts were analyzed by liquid chromatograph-mass spectrometer (LC-MS) analysis, α-mangostin conferring antibacterial property was found in both extracts. The MT-MAE-W/E and MT-Ma-W/E films exhibited 30.22 ± 2.14 and 30.60 ± 2.83 mm of growth inhibition zones against *S. aureus* and 26.50 ± 1.60 and 26.93 ± 3.92 mm of growth inhibition zones against *E. coli*. These clear zones were wider than its crude extract approximately 3 times, possibly because the film formulation enhanced antibacterial activity with sustained release of active compound. Thus, the mangosteen extracts have potential to be used as an antibacterial compound in active packaging.

## 1. Introduction

Thailand is one of the biggest tropical fresh fruit producers. Many kinds of tropical fresh fruits such as mangosteen (*Garcinia mangostana*), rambutan (*Nephelium lappaceum*), and mango (*Mangifera indica*) are exported to other countries and domestically consumed [1,2]. The peels of these fruits are often removed to obtain the edible pulps, rendering the fruit peels by-products. Importantly, these by-products are generated greatly according to the rising amount of consumption and fruit processing which leads to the environmental problems resulting from inappropriate disposal. However, these peels contain useful bioactive compounds, i.e., flavonoids, phenolic acids and polysaccharides [1]. Therefore, these peels exhibit not only antioxidant activity but also antibacterial effects against gram-positive and -negative bacteria [3]. Phenolic compounds found in mangosteen, rambutan and mango peel exhibit significant antibacterial activity [3,4]. Phenolic compounds exhibit antibacterial effect against some gram-positive bacteria, for example: *Staphylococcus aureus*, *Listeria monocytogenes* and *Bacillus subtilis*, and some gram-negative bacteria such as *Salmonella* Typhi, *Escherichia coli* and *Proteus vulgaricus* [3,5]. Although the mechanisms of phenolic compounds on the antibacterial activity are not fully understood, these compounds involve cellular actions. For example, the compounds might modify permeability of cell membrane and change intracellular functions by hydrogen bonding between phenolic molecules and bacterial enzymes [6]. Furthermore, lipophilicity of phenolic compounds enhances their antibacterial activity by interaction with bacterial cell membrane [6]. This result suggests that these fruit peels have the potential to be a source of bioactive compounds which could be used as an antibacterial film for food packaging. Microwave-assisted extraction (MAE) is a novel method developed to be more effective than a conventional method of extraction such as maceration. Previous studies reported that MAE is more effective than a conventional method since this method consumes a small volume of solvent, needs less extraction time and is rapid to transfer energy irradiation which enhances diffusion of solvent, particularly for the extraction of plants containing high-antioxidant compounds [7,8]. Furthermore, microwave radiation activates the heat and pressure, and then leads to changed physical properties of plant cell wall which enhances porosity and permeability [9]. As microbial spoilage is a major problem in food industries due to a decreased shelf-life of foods and illness caused by microbial contamination such as diarrhea and food poisoning, antimicrobial packaging is invented to resolve this issue. Generally, antimicrobial packaging represented as a film has the purpose to extend shelf-life, increase food quality and provide insurance [10,11]. In accordance with this, hydroxypropyl methylcellulose (HPMC), a semi-synthetic cellulose, is used as a film-forming agent for antibacterial edible film [12]. To improve the films’ property in food packaging, antibacterial compounds, i.e., metals, chemicals, enzymes, bacteriocins and plant extracts are incorporated into films. Thymol, cinnamon and clove oils were also used as antibacterial substances to prevent the growth of several pathogens [10]. The ethanolic extract of propolis was used as the active compound incorporated in a pullulan film to obtain antibacterial film coating. The pullulan containing propolis extract exhibited antibacterial activity against *L*. *monocytogenes*, *S. Typhi* and *E. coli* [13]. In this study, the peels of mangosteen, mango and rambutan were extracted by different solvents (water, ethanol 95% and water–ethanol (40:60%)) and methods including maceration and microwave-assisted extraction. The extracts were evaluated for antibacterial activity against *S. aureus* and *E. coli*. The extracts exhibiting lower minimum inhibitory concentration (MIC) were selected to load into HPMC film. Finally, the films loading the extracts were tested for their antibacterial property.

## 2. Results and Discussion

### 2.1. Tropical Fruit Peels Extraction

The extraction yield was shown in Table 1. The highest extraction yield was observed in mango peel extracted by the water and the water–ethanol (40:60%), in maceration and microwave-assisted extractions. Furthermore, during solvent evaporation step, the mango peel extracts in all solvents were thick and viscous, while the dry matter which could be grounded to powder was obtained from the mangosteen and the rambutan extracts. When comparing microwave-assisted extraction (MAE) and maceration using the same solvent obtained from the same fruit peel, the extraction yield was not significantly different (*p* > 0.05). However, the MAE was more effective than maceration in terms of extraction time. Notably, MAE decreased the extraction time from 18 h to only 3 min or 360 times faster.

### 2.2. Phenolic Content of the Fruit Peel Extracts

The phenolic content in the extracts were shown in Figure 1. Mangosteen, mango and rambutan peels contained phenolics in different levels. Overall, rambutan peel extracts exhibited higher phenolic content than other extracts. However, the mangosteen peel extracted in water–ethanol (40:60%) with the MAE method (MT-MAE-W/E extract) had the highest phenolic content (143.58 ± 4.45 mg gallic acid equivalent per 1 g of dry matter (GAE)/g DM). Mango peel extracts exhibited lower phenolic content than other extracts in all solvents and extraction methods. The lowest phenolic content, found in mango peel extract (MG-MAE-W), was only 4.82 ± 0.38 mg GAE/g DM. In previous studies, the phenolic content in mangosteen peel was between 5 and 320 mg GAE/g DM [14,15]. A higher amount of phenolic content obtained from rambutan peel was 100–760 mg GAE/g DM [1], whereas the phenolic content in mango peel depended on cultivars in a range of 0.285–5.742 mg GAE/g DM [16,17]. The MAE tended to be more efficient than the maceration. This could be explained by mechanism of microwave which enhance extraction efficiency due to increasing solvent diffusion and changing permeability of cell wall [9]. The phenolic content in the water and the ethanol extracts were lower than the water–ethanol extract because the mixture of water and ethanol is more effective in recovering antioxidant compounds along with phenolics than the use of pure solvent [18]. The previous study, optimizing the conditions of extracting mangosteen peel by MAE, found that 60% ethanol in water was suitable. It is possible that the addition of water into ethanol increases the polarity and thus reduces the ratio of water in the mixture of water and ethanol, resulting in the higher yield of phenolic content [14]. However, there were other factors that might influence the phenolic content such as pretreatment process, types of solvent, solid–solvent ratio, particle size of sample, extraction time, solvent mixture ratio and variation in plant materials such as season, aging and cultivation.

### 2.3. Xanthones and Flavonoid Content of the Fruit Peel Extracts

Mangosteen, mango and rambutan peels also contained flavonoid compounds. The flavonoid content in the extracts was shown in Figure 2. The highest flavonoid content was obtained from the mangosteen peel extracted in water–ethanol (40:60%) with MAE method (MT-MAE-W/E extract) at 176.84 ± 5.35 mg catechin equivalent (CE)/g DM. Furthermore, in this condition (MAE-W/E), the mango and the rambutan peel extracts exhibited the highest flavonoid content compared between each fruit (37.89 ± 4.23 and 118.26 ± 4.57 mg CE/g DM, respectively). This result can be explained by mechanism of MAE that is proper for extracting antioxidant compounds along with flavonoids since it promotes solvent permeability, cell breaking and dispersing of extracted compound in the solvent [14]. In addition, MT-MAE-W/E extract showed significantly higher flavonoid content (176.84 ± 5.35 mg/g DM) than MT-MAE-W and MT-MAE-E extracts (134.29 ± 8.56 and 112.48 ± 9.72 mg/g DM, respectively) (*p* < 0.05) and the other fruit peels also exhibited similar yields. Clearly, 60% ethanol was a more proper solvent than pure water and ethanol. Other authors found that the hydroethanolic (60%) solvent provided higher yield of flavonoids than aqueous and ethanolic solvent in rambutan peel extraction using ultrasound and boiling [19]. The xanthones (Figure 3a) founded in mangosteen peel were mangostin, garcinone-E, gartanin and smeathxanthone-A, whereas mango peel contains mangiferin and quercetin (Figure 3b) derivatives as major flavonoids. Ellagic acid, corilagin and geraniin were found in rambutan peel [3,20,21]. Similar to other bioactive compounds including phenolics, the flavonoid content was also influenced by not only solvent used and extraction methods, but also other aforementioned factors.

### 2.4. Antibacterial Property of the Fruit Peel Extracts

Antibacterial activity of the extracts against *S. aureus* and *E. coli* investigated by disk diffusion method was shown in Table 2. Full-strength DMSO used as a solvent did not exhibit antibacterial activity. At concentration of 100 mg/mL, rambutan peel extract showed higher antibacterial activity against *S. aureus* than other extracts. For example, RT-Ma-W extract exhibited growth inhibition zone of 11.01 ± 0.25 mm which was significantly higher than MT-MAE-W/E and MG-MAE-W/E extract (9.46 ± 0.20 and 9.63 ± 0.19 mm of growth inhibition zones, respectively) (*p* < 0.05). However, there was no growth inhibition zone of all rambutan peel extracts against *E. coli* at 100 mg/mL. Moreover, all rambutan peel extracts did not exhibit antibacterial activity against both *S. aureus* and *E. coli* at 10 mg/mL. Another study found that rambutan peel extracts in methanol, water and ether at concentration of 2.5 mg/mL exhibited 31.2 mg/mL of minimum inhibitory concentration (MIC) value against *S. aureus* but these extracts did show growth inhibition zones against *E. coli* [22]. For mango peel extracts, the growth inhibition zone was not observed in MG-MAE-W and MG-Ma-W extracts in both 100 mg/mL and 10 mg/mL. In addition, all of mango extracts did not show growth inhibition zones at 10 mg/mL against *E. coli*. It is possible that the amount of phenolics and flavonoids in mango peel extracts which possess antibacterial property was not sufficient for bacterial killing. For the water (W) extraction of mangosteen peel, MT-MAE-W and MT-Ma-W extracts did not show growth inhibition zones against both *S. aureus* and *E. coli*. However, the other solvent extractions (ethanol and water–ethanol), MT-MAE-E, MT-MAE-W/E, MT-Ma-E and MT-Ma-W/E exhibited growth inhibition zones against *S. aureus* and *E. coli* at concentration of 10 and 100 mg/mL of extracts. This thus indicated that theses extracts possessed lower MIC when compared to the others. Interestingly, extraction with water–ethanol (40:60%) resulted in a wider growth inhibition zones against both bacteria than ethanolic extraction significantly (*p* < 0.05) at concentration of 100 mg/mL. For example, MT-MAE-W/E displayed 9.46 ± 0.20 and 9.88 ± 0.21 mm of the clear zone, while MT-MAE-E presented 9.19 ± 0.34 and 8.94 ± 0.39 mm of the clear zone against *S. aureus* and *E. coli*, respectively. This finding may be due to a higher phenolic and flavonoid contents than the other solvents when the extraction with water–ethanol was executed. Both phenolic compounds and flavonoids display an antimicrobial property, but the mechanisms of bacterial inhibition are different. Phenolic compounds act on bacterial cells by an interference of bacterial membrane that leads to an increased membrane permeability and losing ion, ATP and other cytoplasm contents [23]. Flavonoids including mangostin inhibit bacterial growth by different mechanisms such as inhibition of nucleic acid synthesis, inhibition of energy metabolism, inhibition of the porin on the cell membrane and alteration of the membrane permeability [24]. From these results, MT-MAE-W/E and MT-Ma-W/E extracts were selected to analyze by LC-MS and loaded into HPMC film to obtain active antibacterial materials.

### 2.5. Liquid Chromatography-Mass Spectrometry (LC-MS) Analysis of Selected Extracts

LC-MS analysis revealed that there were more than 300 compounds found from scanning in MT-MAE-W/E and MT-Ma-W/E crude extracts. Full LC-MS chromatograms of MT-MAE-W/E (a) and MT-Ma-W/E (b) in negative ion mode of analysis were shown in Appendix A. Notably, α-mangostin was found in both extracts. Retention time of α-mangostin was 29.26 and 29.26 min in negative data analysis of MT-MAE-W/E and MT-Ma-W/E, respectively (Appendix A). At this retention time, the data were matched with α-mangostin (C_24_H_26_O_6_) with 99.51% matching score mass per chart ratio of 409.16 in the software library. In addition, α-mangostin was widely known as the dominant bioactive compound in mangosteen peel and found in LC-MS analysis [25,26]. These results confirmed that α-mangostin was found in both extracts. The mass spectra of α-mangostin and its structure are shown in Figure 4 and Figure 5, respectively. Additionally, α-mangostin is a major compound in mangosteen peel (pericarp). Previous studies found that most of the biological activity of mangosteen peel is associated with α-mangostin [25,27]. Interestingly, α-mangostin extracted from mangosteen peel demonstrated a broad range of physiological activity including antioxidation, antiproliferation, anticancer and antimicrobial activity [27]. Overall, mass-spectra of both extracts exhibited similar pattern, but the mass-spectra of MT-Ma-W/E showed more peaks than MT-MAE-W/E with stronger intensity. This result indicated that different extraction methods influenced extracted substances. The other peaks in the mass-spectra were not relevant to phenolic, flavonoid and other known compounds in mangosteen peel thus they might be an impurity co-isolated during the extraction procedures. For instance, a peak at 256.24 *m*/*z* in both extracts might indicate the presence of butyl dodecanoate (C_16_H_32_O_6_) with 86.60% matching score. Other peaks such as a strong peak at 520, 610 and 790 *m*/*z* were not yet identified.

### 2.6. Characteristics of the Films

Weight, thickness, diameter and gross appearance of films are shown in Table 3 and Figure 6. The thickness of the MT-MAE-W/E film (0.852 ± 0.028 mm) was not significantly different from the MT-Ma-W/E film (0.848 ± 0.037 mm). Similarly, the significant difference between the weight of MT-MAE-W/E film (0.050 ± 0.003 g) and MT-Ma-W/E film (0.047 ± 0.005 g) was not observed. However, the addition of MT-MAE-W/E and MT-Ma-W/E extracts into the films clearly increased the thickness and added the weight into the blank film. After the drying process, the solvent was evaporated and the extract was retained in the film matrix as indicated by the final film color showing that the extract constituted the film.

### 2.7. Mechanical Property of the Films

The mechanical properties of the films containing extracts are shown in Table 4. In all parameters, MT-MAE-W/E did not significantly exhibit different values of tensile strength, elongation at break and Young’s modulus (2.71 ± 0.30 N/mm^2^, 0.68 ± 0.10% and 20.48 ± 1.57 N/mm^2^, respectively) when compared with MT-Ma-W/E film (2.64 ± 0.42 N/mm^2^, 0.60 ± 0.06% and 20.44 ± 1.28 N/mm^2^, respectively). However, when extracts were added into the film, the mechanical properties were changed. The tensile strength and Young’s modulus of the film were decreased from 4.79 ± 1.57 to 2.71 ± 0.30 N/mm^2^ and 25.40 ± 2.75 to 20.48 ± 1.57 N/mm^2^, when compared between blank film and the MT-MAE-W/E film, respectively. The tensile strength is associated with the hardness and stickiness of material and the Young’s modulus is related to material rigidity [28,29]. These results indicated that the MT-MAE-W/E film and MT-Ma-W/E film were softer than the blank film. In addition, the elongation at break was reduced from 8.72 ± 2.09 to 0.68 ± 0.10% in the MT-MAE-W/E film when compared with the blank film. The elongation at break is related to the ability to deform under applied force [28]. This result indicated that the ability of the films to elongate was reduced when the MT-MAE-W/E and MT-Ma-W/E were added into the films. The change in the mechanical property might result from the rearrangement of the polymer network. When substances are incorporated into polymer, they interfere the polymer–polymer interaction and lead to the rearrangement of the polymer network that alters the mechanical properties of materials. The addition of extracts into the films might possibly develop a structural discontinuity, thereby producing a softer film with reduced elongation property [30].

### 2.8. Antibacterial Property of Films Containing Extracts

Agar disk diffusion method was performed to investigate an antibacterial activity of HPMC films containing mangosteen extract against *S. aureus* and *E. coli*. In previous study, HMPC films containing essential oil were evaluated for antibacterial activity by the agar disk diffusion method [31]. The positive control was a clindamycin HCl solution equivalent to 1% clindamycin base and the negative control was HPMC film without the extracts. The growth inhibition zones of the films are shown in Table 5. Consistent with previous reports, blank film did not show growth inhibition zone because HPMC did not exhibit antibacterial activity against both gram positive and negative bacteria [32,33,34]. Compared between 2 extracts, the MT-MAE-W/E film displayed 30.22 ± 2.14 mm of growth inhibition zone against *S. aureus* which was not statistically different (*p* > 0.05) from the MT-Ma-W/E film (30.60 ± 2.83 mm of growth inhibition zone). Likewise, significant difference of antibacterial activity against *E. coli* was not found between the MT-MAE-W/E film (26.50 ± 1.60 mm of growth inhibition zone) and the MT-Ma-W/E film (26.93 ± 3.92 mm of growth inhibition zone). At 100 mg/mL, the MT-MAE-W/E and MT-Ma-W/E crude extracts insignificantly exhibited the growth inhibition zones against both *S. aureus* and *E. coli*. The films containing these extracts (12% *w*/*v*) exhibited clear zone against *S. aureus* and *E. coli*., while clindamycin showed greater inhibition of *S. aureus* than *E. coli* growth confirming a selective killing activity by clindamycin against gram-positive microbes [35]. Interestingly, film preparation enhanced antibacterial activity of the extracts. The growth inhibition zone against *S. aureus* of MT-MAE-W/E extract clearly increased from 9.46 ± 0.20 to 30.22 ± 2.14 mm, when it was loaded in the HPMC film. The film formulation increased contact time and may provide sustained release of the extracts. When a hydrophilic film, i.e., HPMC film makes contact with water, the water molecules penetrate into polymer matrix resulting in film swelling, enlarging of polymer chains, and releasing active compounds into surrounding area [36]. Several active packaging films were developed to obtain a sustained release by loading active compounds in as much as the gradual release is a crucial characteristic required for the films’ functionality [37]. For example, MC/HPMC-composited film containing propolis extract showed a prolonged release which can inhibit bacterial growth for four weeks [38]. In another study, chitosan film containing cinnamon oil exhibited sustained release of active compound to control bacterial growth [39]. Thus, the films containing mangosteen extracts in this study might also gradually re-lease bioactive compounds in which further examination may corroborate this speculation.

## 3. Materials and Methods

### 3.1. Materials

Hydroxypropyl methylcellulose E15 (HPMC E15; AnyCoat-C AN15^®^) was purchased from Lotte Fine Chemical (Ulsan, Korea). Folin–Ciocalteau phenol reagent, gallic acid monohydrate ACS reagent (≥98.0%) and tryptic soy broth were purchased from Sigma-Aldrich (Saint Louis, MO, USA). Sodium bicarbonate (NaHCO_3_) was purchased from Merck (Darmstadt, Germany). Sodium nitrate (NaNO_3_) and aluminum chloride (AlCl_3_) were purchased from Merck (Darmstadt, Germany). Sodium hydroxide (NaOH) was purchased from Ajax Finechem (Victoria, Australia). A brain heart infusion (BHI) broth powder was purchased from HiMedia (Mumbai, India). Clindamycin solution; Clinda-M, RPC^®^ (equivalent to 1% clindamycin base) was purchased from RPC International Co., Ltd. (Bangkok, Thailand).

### 3.2. Tropical Fruits Extraction

#### 3.2.1. Tropical Fruit Peel Collection

Mangosteen (*Garcinia mangostana* Linn.), mango (*Mangifera indica* cv. Nam Dokmai) and rambutan (*Nephelium lappaceum* Linn.) were collected from a local market in Chiang Mai, Thailand. The peels were removed and cut to small pieces. Then, the peels were dried in hot air oven (B40 Memmert, Schwabach, Germany) at 45 ± 2 °C for 24 h. After that, the dried peels were ground to powder and sieved (0.6 mm diameter).

#### 3.2.2. Microwave-Assisted Extraction

Each fruit peel powder (1 g) was mixed separately with water–ethanol mixture (20 mL) in water, ethanol 95% and water–ethanol (40:60%). Then the mixture was placed in the phase-power control microwave extraction (PC-MHG) system. The extraction process was operated at a frequency of 2450 MHz at 200 W for 3 min. Insoluble particles were removed by filtrating with vacuum at room temperature. The filtrate was concentrated by rotary evaporator resembled over water bath at 50 °C (Rotavapor^®^ R-300, Flawil, Switzerland) to obtain a thick or dry matter. [3,40,41]. The extraction was performed 3 times and the final extraction yield was calculated using an Equation (1).
(1)$$\mathrm{Extraction}\;\mathrm{yield}\;\left(\%\right)\;=\;\;\frac{weight\;of\;crude\;extract\left(g\right)}{weight\;of\;peel\;powder\;\left(g\right)}\;\times 100$$

#### 3.2.3. Maceration

Mango, mangosteen or rambutan peel powders were saturated separately in water, ethanol 95% and water–ethanol (40:60%) for 18 h in a dark room at room temperature. The powder–solvent ratio was 1 g/20 mL. The macerate was filtrated under vacuum at room temperature to remove insoluble particles, then the filtrate was concentrated by rotary evaporator resembled over water bath at 50 °C (Rotavapor^®^ R-300, Flawil, Switzerland) until a thick or dry extract was obtained. [3,42]. The extraction of each plant was replicated 3 times. The extraction yield was calculated using Equation (1), as described in Section 3.2.2.

### 3.3. Phenolic Content Analysis

Folin–Ciocalteau procedure, method to determine phenolic compounds, was performed as previously described [43]. The extracts were diluted 100–1000 times by each extracted solvent. The dilute extracts (30 µL) were added into the 96-well plate. Then, 60 µL of 10% *v*/*v* Folin–Ciocalteau reagent was added. The plate was allowed to stand 1 min and 210 µL of 6% *w*/*v* NaHCO_3_ was added, respectively. Then, the plates were placed in the dark at room temperature for 90 min. After that, the phenolic content was determined by the absorbance microplate reader (Spectro star Nano, BMG Labtech, Ortenberg, Germany) at 725 nm wavelength. Gallic acid was used as a standard for standard curve in ranging of 0–200 mg/mL. The phenolic content was calculated in term of gram of gallic acid equivalent per gram of sample.

### 3.4. Flavonoid Content Analysis

Flavonoid content in the extracts was quantitated by a modified colorimetric method as described earlier [44]. The samples were diluted by the same solvent used for extraction. Twenty-five micro liters of sample was mixed with 125 µL of distilled water. Then, 5% *w/v* NaNO_3_ 7.5 µL was added with 5 min holding and then 10% *w/v* AlCl_3_ was added with allowing to stand at room temperature for 6 min. Subsequently, 1 N NaOH was added into each plate immediately. Finally, distilled water was added, and the samples were analyzed for flavonoid content by the absorbance microplate reader (Spectro star Nano, BMG Labtech, Ortenberg, Germany) at 510 nm reading. The standard curve was made from catechin in different concentration ranging from 0 to 300 µg/mL. The flavonoid content was presented in term of catechin equivalent per gram of sample.

### 3.5. Antibacterial Activity Test of Extracts

The extracts were evaluated for antibacterial activity against *S. aureus* and *E. coli*. The method was adapted from [45]. Briefly, stock cultures of *S. aureus* (DMST 8840) and *Escherichia coli* (O157:H7 DMST 12743) were inoculated on BHI agar (HiMedia Laboratories, Mumbai, India) and grown at 37 °C for 24 h. They were subsequently inoculated into BHI broth and grown at 37 °C under an aerobic atmosphere. After 24 h of incubation, the optical density of the test bacteria was determined by a UV spectrophotometer (Beckman Coulter, Fullerton, CA, USA) at 600 nm (OD600). Whatman^®^ antibiotic assay discs (GE Healthcare, Pittsburgh, PA, USA) with 6 mm diameter were loaded 20 µL extracts, dissolved in dimethyl sulfoxide (DMSO, Amresco, Solon, OH, USA) to achieve the concentration of 10 and 100 mg/mL. Then, 100 µL of working bacterial stock (OD600 = 0.1) was plated on a sterile BHI agar and allowed to dry. Bacterial agars with tested materials were incubated at 37 °C under an aerobic condition until the bacterial lawn was observed. The antibacterial activity was reported at 24 h after incubation in term of growth inhibition zones, measured by Mitutoyo^®^ Digimatic caliper (Mitutoyo Corporation, Kanagawa, Japan). The disk loaded with clindamycin solution equivalent to 1% clindamycin base was used as a positive control. Two tested extracts displaying the highest antibacterial activity against *S. aureus* and *E. coli* were chosen for identifying compounds in the next experiment and further loading into a film.

### 3.6. Compounds Identification

#### 3.6.1. Sample Preparation

To eradicate lipid and pigment, sample (3 mg) was dissolved in 1 mL of water–ethanol (40:60%), then transferred to dispersive solid phase extraction (Agilent Bond Elut, Santa Clara, CA, USA) and centrifuged at 1000× *g* rpm for 15 min. The supernatant was collected and diluted to 1 ppm by water–ethanol (40:60%), respectively.

#### 3.6.2. LC-MS Analysis

To identify compounds which might exhibit antibacterial property, the extracts were analyzed using UHPLC (Agilent 1290 Infinity II LC System, Santa Clara, CA, USA) coupled with quadrupole time of fight mass spectrophotometer (Agilent 6546 LC/Q-TOF, CA, USA). The reversed-phase column (Agilent Eclipse Plus C18 RRHD 2.1 × 150 mm, 1.8 µm, Santa Clara, CA, USA) was employed in the UHPLC system and the column temperature was set at 35 °C. Gradient dilution, that was composed of water containing 0.1% formic acid (A) and acetonitrile (B), with 0.3 mL/min of flow rate was performed as follows: 0 min, 5% B; 0–5 min, 20% B; 5–10 min, 30% B, 10–15 min, 35% B, 15–20 min, 45% B, 20–25 min, 75% B; 25–45 min, 95% B; and 45–50 min, 95% B. The injection volume was 2 µL and condition of auto-sampler was set at room temperature. 

Mass spectrometric detector was used with dual Agilent Jet Stream Electrospray Ionization (Dual AJS ESI) for detection. The capillary voltage was set as 3.5 kV. The source temperature was maintained at 125 °C. The collision gas was N_2_ from gas generator (PEAK Scientific genius XE nitrogen, Scotland, UK), desolvating at 350 °C. The flow rate of cone gas was set as 11 L/min and nebulizer pressure was 35 psi. Both positive and negative data were obtained form same condition and integrated at 0.5 s of scanning time. The raw data were analyzed by Agilent Mass hunter qualitative software analysis version 10.0 (Agilent, Santa Clara, CA, USA) with Metlin library. The parameter included a retention time in a range of 0.5–50 min and a mass range of 100–1000 Da. To ensure the presence of the expected substances, identified bioactive compounds were matched to the database in the library as well as the data obtained from previous study, and at least 90% matching score must be reported.

### 3.7. Preparation of the Film Containing Extract

Film containing extract was prepared by solvent-casting method. Hydroxypropyl methylcellulose (HPMC) 20 g was dissolved in 100 mL of water–ethanol (40:60%) at room temperature. Glycerol which was used as plasticizer was added into the polymer solution in a ratio of 40% *w*/*w* based on dry HPMC. Mangosteen peel extract (1.2 g) was redissolved separately in 5 mL of water–ethanol (40:60%). Then, 5 mL of HPMC solution was mixed homogeneously with the extract solution at room temperature by a magnetic stirrer. The homogenous mixture was poured into a petri dish (6 cm in diameter), dried at room temperature for 24 h and the film was removed from the petri dish. Finally, the film was cut into circles of diameter 7.94 mm by a hollow hole punch cutter.

### 3.8. Characterization of the Films

The films were visually observed as gross appearance. The thickness of films was measured by thickness gauge measurement (GT-313-A, Gotech testing machines Inc., Taichung, Taiwan). The weight of films was determined by analytical balance (Pioneer^®^, Ohaus, Parsippany, NJ, USA). The film was cut to circle shape by molding cutter (7.94 mm in diameter) and the film diameter was measured by the Digimatic caliper.

### 3.9. Mechanical Property Test

The films were cut to circle with 15 mm diameter. The films were tested for tensile properties by Texture analyzer TX.TA plus (Stable Micro Systems, Surrey, UK) with 5 kg load cell (0.001 N of sensitivity). The plane flat-faced surface probe (2 mm of diameter) was used to compress the films. The distance between the films and the probe was 10 mm. The probe’s speed was set as 1 mm/s. Data such as time (s), force (N) and distance (mm) was recoded when the probed contacted with the films. Each film sample was tested in replicate independently 6 times. The mechanical properties were presented in term of tensile strength, elongation at break and Young’s modulus [29,46].

### 3.10. Antibacterial Activity of Films Containing Extracts

The antibacterial property of films containing the active extracts was evaluated by disk diffusion method which described [45,46]. The process to prepare bacteria culture was described in Section 3.5. The films were cut into circle shape with 7.94 mm in diameter. The blank film was used as negative control. The Whatman^®^ antibiotic assay discs (7.94 mm in diameter) containing 1% clindamycin solution was sued as the positive control.

### 3.11. Statistic Analysis

Data were presented as mean ± standard deviation (S.D.). SPSS software (version 17; IBM corporation, Armonk, NY, USA) was used to analyze the statistical significance where a significance level (*p*-value) of less than 0.05 was considered statistically different.

## 4. Conclusions

Crude extracts of mangosteen, rambutan and mango peels exhibited different percentage yield, flavonoids, phenolics and antibacterial property. Furthermore, different extraction method and solvent influenced these characteristics. Mango extracted by the MAE and the maceration showed higher yield but its crude extracts were sticky which may indicate a higher sugar content. In terms of bioactive compounds, MAE tended to yield higher phenolic and flavonoid content than maceration since microwave mechanism enhanced efficiency to extract phenolics and flavonoids from plants. Water–ethanol (40:60%) provided higher phenolics and flavonoids than other solvents since its polarity might be suitable to extract the bioactive compounds. Mangosteen extracts, with the exception of MT-MAE-W and MT-Ma-W, exhibited higher antibacterial activity against *S aureus* and *E. coli*. Overall, MT-MAE-W/E and MT-Ma-W/E extracts were more suitable to use as a bioactive substance in active packaging owing to higher phenolic and flavonoid contents as well as their greater antibacterial activity against *S. aureus* and *E. coli* in lower concentration (10 mg/mL). Furthermore, α-mangostin, a known bioactive compound possessing an antimicrobial activity, was found in MT-MAE-W/E and MT-Ma-W/E extracts in LC-MS analysis. Despite the reduced mechanical properties of the film containing mangosteen extracts, the films showed a promising result in killing microbes potentially contaminating packaged food. MT-MAE-W/E and MT-Ma-W/E films exhibited roughly 3 times wider growth inhibition zones against *S. aureus* and *E. coli* than MT-MAE-W/E and MT-Ma-W/E crude extracts. The film formulation enhanced antibacterial activity due to sustained release of the extract from film matrix. This study indicated that tropical fruit extracts, especially mangosteen extracted in water–ethanol (40:60%) had the potential to be an alternative substance used for antibacterial active film packaging. However, a further optimization of the film is required to obtain an optimal condition for the films used in the food packaging industry. The findings from this study may be useful for further studies to purify the extract in order to use the natural active compound in film formulation for active antibacterial packaging.

## Figures and Tables

**Figure 1 molecules-26-02265-f001:**
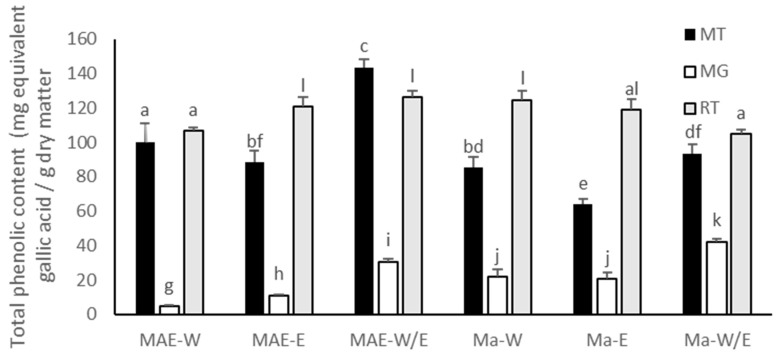
Total phenolic content of extracts in mg equivalent gallic acid/g dry matter: mangosteen peel (MT); mango peel (MG); rambutan peel (RT); microwave-assisted extraction (MAE); maceration (Ma); water (W); ethanol 95% (E); water–ethanol (40:60%) (W/E). The data was performed in term of Mean ± S.D. Different letters (a–l) indicate a significant difference (*p* < 0.05) between each bar.

**Figure 2 molecules-26-02265-f002:**
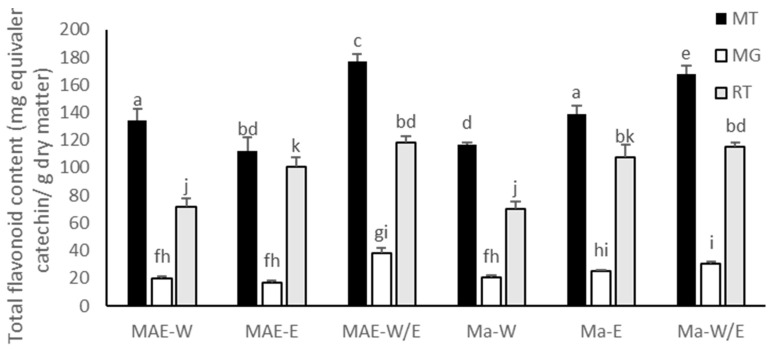
Total flavonoid content of extracts in mg equivalent catechin/g dry matter: mangosteen peel (MT); mango peel (MG); rambutan peel (RT); microwave-assisted extraction (MAE); maceration (Ma); water (W); ethanol 95% (E); water–ethanol (40:60%) (W/E). The data are presented in terms of Mean ± S.D. Different letters (a–k) indicate a significant difference (*p* < 0.05) between each bar.

**Figure 3 molecules-26-02265-f003:**
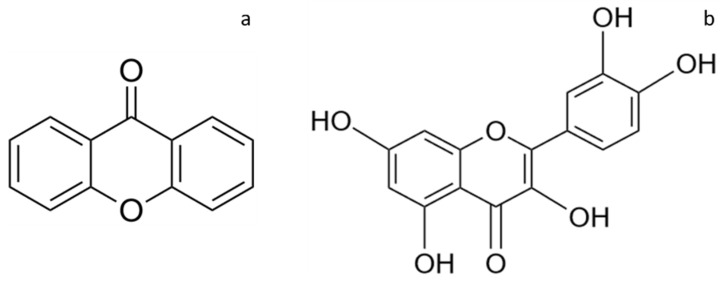
Chemical structures of xanthone (**a**) and quercetin (**b**).

**Figure 4 molecules-26-02265-f004:**
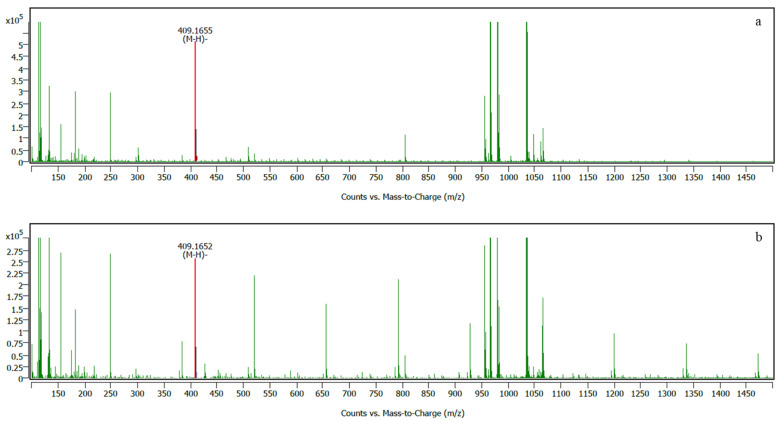
Mass spectrum of MT-MAE-W/E (**a**) and MT-Ma-W/E (**b**) extracts showing α-mangostin (red mark).

**Figure 5 molecules-26-02265-f005:**
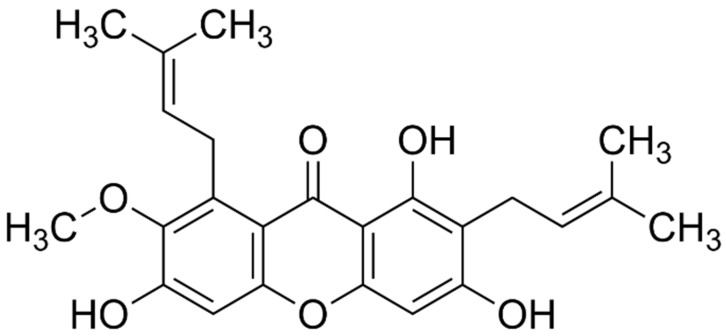
Chemical structure of α-mangostin.

**Figure 6 molecules-26-02265-f006:**
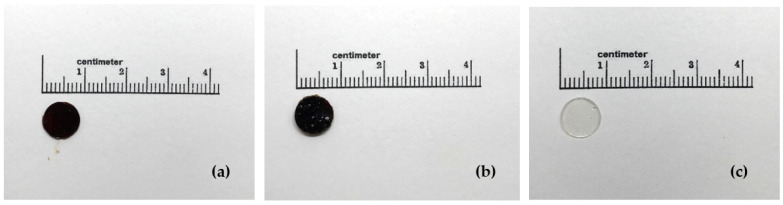
Gross appearance of MT-MAE-W/E (**a**), MT-Ma-W/E (**b**) and blank (**c**) films.

**Table 1 molecules-26-02265-t001:** The extraction yield (%) of each fruit peel.

Fruit Peel	Extraction Method	Solvent	Yield (%)
Mangosteen		Water	17.02 ± 2.40 ^a^
	Maceration	Ethanol 95%	16.98 ± 1.90 ^a^
		Water–ethanol (40:60)	24.98 ± 1.78 ^bc^
	MAE	Water	17.32 ± 0.17 ^a^
		Ethanol 95%	16.42 ± 3.08 ^a^
		Water–ethanol (40:60)	22.84 ± 1.24 ^bc^
Mango		Water	48.60 ± 0.28 ^d^
	Maceration	Ethanol 95%	25.84 ± 1.81 ^bc^
		Water–ethanol (40:60)	46.08 ± 0.34 ^e^
	MAE	Water	45.32 ± 4.41 ^de^
		Ethanol 95%	26.68 ± 0.17 ^b^
		Water–ethanol (40:60)	44.92 ± 0.87 ^de^
Rambutan		Water	26.68 ± 1.07 ^b^
	Maceration	Ethanol 95%	26.92 ± 1.41 ^b^
		Water–ethanol (40:60)	34.92 ± 0.96 ^f^
		Water	28.05 ± 2.60 ^b^
	MAE	Ethanol 95%	27.36 ± 2.26 ^b^
		Water–ethanol (40:60)	34.58 ± 2.97 ^f^

The experiment was performed in triplicate. Percentages of yield were showed as mean ± S.D. Values in same column with different letters (^a–f^) indicate a significant difference (*p* < 0.05).

**Table 2 molecules-26-02265-t002:** Growth inhibition zones of extracts against *S. aureus* and *E. coli.*

Extract	Growth Inhibition Zone (mm)			
	100 mg/mL		10 mg/mL	
	*S. aureus*	*E. coli*	*S. aureus*	*E. coli*
MT-MAE-W	NA	*NA*	*NA*	*NA*
MT-MAE-E	9.19 ± 0.34 ^c1^	8.94 ± 0.39 ^bcf#^	8.50 ± 0.11 ^b1^	9.18 ± 0.05 ^b#^
MT-MAE-W/E	9.46 ± 0.20 ^aj1^	9.88 ± 0.21 ^ag#^	8.07 ± 0.08 ^a2^	8.89 ± 0.16 ^a^*
MT-Ma-W	NA	NA	NA	NA
MT-Ma-E	9.54 ± 0.25 ^acej1^	9.24 ± 0.17 ^bd#^	8.36 ± 0.06 ^ab2^	8.00 ± 0.14 ^d^*
MT-Ma-W/E	9.65 ± 0.07 ^acd1^	9.22 ± 0.41 ^abf#^	8.24 ± 0.47 ^ab1^	8.29 ± 0.06 ^c^*
MG-MAE-W	NA	NA	NA	NA
MG-MAE-E	9.41 ± 0.25 ^acfj1^	7.67 ± 0.38 ^e^	8.33 ± 0.35 ^a2^	NA
MG-MAE-W/E	9.63 ± 0.19 ^acj1^	8.83 ± 0.50 ^bf^	8.69 ± 0.25 ^b2^	NA
MG-Ma-W	NA	NA	NA	NA
MG-Ma-E	9.83 ± 0.46 ^agj1^	9.35 ± 0.65 ^bfg^	9.67 ± 0.46 ^c1^	NA
MG-Ma-W/E	9.94 ± 0.32 ^bdegj1^	8.55 ± 0.29 ^cdf^	9.17 ± 0.06 ^c2^	NA
RT-MAE-W	7.95 ± 0.32 ^h^	NA	NA	NA
RT-MAE-E	10.68 ± 0.16 ^bi^	NA	NA	NA
RT-MAE-W/E	10.29 ± 0.25 ^bg^	NA	NA	NA
RT-Ma-W	11.01 ± 0.25 ^i^	NA	NA	NA
RT-Ma-E	10.10 ± 0.66 ^bdg^	NA	NA	NA
RT-Ma-W/E	10.07 ± 0.33 ^bdfgj^	NA	NA	NA
Clindamycin HCl solution (1%)	37.228 ± 0.564 ^k1^	32.092 ± 0.483 ^h#^	37.043 ± 0.606 ^d1^	32.666 ± 0.419 ^e#^
100% DMSO	NA	NA	NA	NA

The experiment was performed in triplicate. Mean ± S.D. values in the same column with different letters (^a–j^) indicate a significant difference (*p* < 0.05). Values in the same row with different numbers (^1,2^) indicate a significant difference (*p* < 0.05) of the growth inhibition zones against *S. aureus* in different concentration. Values in the same row with different symbols (^#^,*) indicate a significant difference (*p* < 0.05) of the growth inhibition zones against *E. coli* in different concentrations.

**Table 3 molecules-26-02265-t003:** Weight, thickness and diameter of film samples.

Film Sample	Thickness (mm ± SD)	Weight (g ± SD)	Diameter (mm)
MT-MAE-W/E film	0.852 ± 0.028 ^a^	0.050 ± 0.003 ^a^	7.94
MT-Ma-W/E film	0.848 ± 0.037 ^a^	0.047 ± 0.005 ^a^	7.94
Blank film	0.335 ± 0.024 ^b^	0.021 ± 0.001 ^b^	7.94

For each test, means with the same letter are not significantly different. Thus, means with different letters, e.g., “^a^” or “^b^”, are statistically different (*p* < 0.05).

**Table 4 molecules-26-02265-t004:** Mechanical properties of prepared films.

Film	Tensile Strength (N/mm^2^)	Elongation at Break (%)	Young’s Modulus (N/mm^2^)
MT-MAE-W/E film	2.71 ± 0.30 ^a^	0.68 ± 0.10 ^a^	20.48 ± 1.57 ^a^
MT-Ma-W/E film	2.64 ± 0.42 ^a^	0.60 ± 0.06 ^a^	20.44 ± 1.28 ^a^
Blank film	4.79 ± 1.57 ^b^	8.72 ± 2.09 ^b^	25.40 ± 2.75 ^b^

For each test, means with the same letter are not significantly different. Thus, means with the different letter, e.g., ‘^a^’ or ‘^b^’, are statistically different (*p* < 0.05).

**Table 5 molecules-26-02265-t005:** Growth inhibition zones of films containing extracts against *S. aureus* and *E. coli.*

Samples	Growth Inhibition Zone (mm)	
	*S. aureus*	*E. coli*
Blank film	NZ	NZ
MT- MAE-W/E film	30.22 ± 2.14 ^a1^	26.50 ± 1.60 ^a1^
MT- Ma-W/E film	30.60 ± 2.83 ^a1^	26.93 ± 3.92 ^a1^
Clindamycin HCl solution	34.76 ± 0.10 ^b1^	16.58 ± 0.60 ^b2^

The experiment was performed in triplicate. Mean ± S.D. values in the same column with different letters (^a,b^) indicate a significant difference (*p* < 0.05). Values in the same row with different numbers (^1,2^) indicate a significant difference (*p* < 0.05). NZ indicates no growth inhibition zone detected.

## Data Availability

Data is contained within the research article.

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
