# Peer review of "Extraction of Tropical Fruit Peels and Development of HPMC Film Containing the Extracts as an Active Antibacterial Packaging Material"

_molecules, 2021, doi:10.3390/molecules26082265_

Round 1
Reviewer 1 Report
The manuscript, "Extraction of Tropical Fruit Peels and Development of HPMC Film Containing the Extracts as an Active Antibacterial Packaging Material", reports the study of the crude extract from peels of mangosteen, mango, and rambutan. The extracts were obtained by different solvents and microwave-assisted extraction (MAE). In my opinion, the manuscript is not suitable for publication in Molecules. There are papers that describe the occurrence of phenolic compounds in peels of tropical fruit, including mangosteen and mango. Besides, mango by-products (kernel and peel) were also used in food packaging, (Review: Foods 2020, 9, 857; doi:10.3390/foods9070857). Other comments and questions are the following:
- In the abstract, “In liquid chromatograph-mass spectrometer (LC-MS) analysis, α-mangostin conferring antibacterial property was found in both extracts”, but only extracts were tested.
- In table 2, the value inhibitory clear zone of positive control (Clindamycin HCl solution) was not shown.
- Why did the authors not explore the different methods for extraction? Would not be interesting a qualitative and quantitative analysis for the extracts? These results could help to monitor the chemical variability (mangostin or other phenolic compounds) in the composition of the peels of tropical fruits, since this variability changes the concentration of the substances responsible for the activity and, consequently, the antibacterial activity.
- If the main goal was to find a compound or extract with antibacterial activities to improve the films’ property in food packaging. Wouldn't it be better also to test the film against one or more bacteria as Salmonella sp., Campylobacter sp, Bacillus sp., Clostridium sp., or Vibrio sp.?
- LC / MS could be added to the supplementary material or other data from the analyzes.
- Were chemical (UV absorbing), physical or mechanical properties of the film tested? Or will be?
Reviewer 2 Report
The manuscript "Extraction of Tropical Fruit Peels and Development of HPMC Film Containing the Extracts as an Active Antibacterial Packaging Material" is interesting and small corrections could improve it.
Detailed comments:
line 57-59 - for the purpose of introducing abbreviations of the names of microorganisms, it is generally accepted that the Latin name is first spelled in full, and the abbreviation continues in the text.
line 58 - It should be Salmonella Typhi
line 73-75 - an example of an additive to edible films is also a propolis extract with a high polyphenol content (i.e. doi: 10.1111 / ijfs.14753, https://doi.org/10.1007/s11947-020-02487-w)
line 194 - the name "growth inhibition zones" is used more often than "inhibitory clear zone"
line 260-262 - please specify the diameter of the film disc in the description
line 325 - should be antibacterial instead of anti-bacterial
line 327 - please provide one collection number - ATCC 25923 or DMST 8840
Author Response
Response Letter
Manuscript ID: molecules-1178441
Title: Extraction of Tropical Fruit Peels and Development of HPMC Film Containing the Extracts as an Active Antibacterial Packaging Material
Special issue: Circular Economy: Agri-Food Byproducts as Source of Bioactive Compounds
Comments of reviewer 2
The manuscript "Extraction of Tropical Fruit Peels and Development of HPMC Film Containing the Extracts as an Active Antibacterial Packaging Material" is interesting and small corrections could improve it.
Response: The authors thank the reviewer 2 for help to improve the manuscript and we try to answer for all questions.
Detailed comments:
1. line 57-59 - for the purpose of introducing abbreviations of the names of microorganisms, it is generally accepted that the Latin name is first spelled in full, and the abbreviation continues in the text.
Response: The abbreviation names of bacteria in the introduction were deleted, and only full names remain (please see line 56 -57).
2. line 58 - It should be Salmonella Typhi
Response: Salmonella typhi was changed to Salmonella Typhi. (please see line 57).
3. line 73-75 - an example of an additive to edible films is also a propolis extract with a high polyphenol content (i.e. doi: 10.1111 / ijfs.14753, https://doi.org/10.1007/s11947-020-02487-w)
Response: The example of the use of propolis extract as the active compound in coating film for antibacterial application was added (please see line 82 – 84 and ref 13).
4. line 194 - the name "growth inhibition zones" is used more often than "inhibitory clear zone"
Response: The word “inhibitory clear zone” in the manuscript was changed to "growth inhibition zones" (please see the revised manuscript).
5. line 260-262 - please specify the diameter of the film disc in the description
Response: The film diameter was described in method (please line 438).
6.line 325 - should be antibacterial instead of anti-bacterial
Response: The word “anti-bacterial” was changed to “antibacterial” (please see line 363).
7. line 327 - please provide one collection number - ATCC 25923 or DMST 8840
Response: We have edited to S. aureus DMST8840. Sorry for the confusion (please see line 364).
Reviewer 3 Report
This manuscript entitled “Extraction of Tropical Fruit Peels and Development of HPMC Film Containing the Extracts as an Active Antibacterial Packaging Material”. In this study, the peels of mangosteen (Garcinia mangostana), rambutan (Nephelium lappaceum), and mango (Mangifera indica) were extracted to obtain bioactive compound by microwave-assisted extraction (MAE) and maceration with water, ethanol 95% and ethanol/water. Meanwhile, the effect of antibacterial of HPMC film containing three tropical fruit peels extractions was researched on Staphylococcus aureus (S. aureus) and Escherichia coli (E. coli). The result of research shows that the mangosteen extracts have a potential to be used as an antibacterial compound in an active packaging. The work is meaningful but some detailed comments are made as below:
- Line 45-52, add references.
- There is some confusion in the abbreviation in the article. It is suggested to choose a suitable abbreviation and keep the whole text unified. Like Mi is the same with microwave-assisted extraction (MAE) from my perspective.
- Delete the sentence in line 88-92, consider its relevance to the topic.
- The p value was mentioned in line 96 and line 144, but the results of statistical analysis are not shown in the table 1 and Figure 2.
- Line 115-121, it is suggested to put two sentences in the introduction section. Although microwave-assisted extraction (MAE) has advantages over traditional methods, the advantages of MAE in extracting phenolic acids cannot be directly proved especially this part is the result of phenolic content.
- It is recommended that the figures are further optimized to ensure its resolution, such as Figure 1, 2 and 3.
- Results descriptive and inferential sentences should be used with caution, such as lin177-184. Please review the full text.
- It is recommended to add chromatography (HPLC, UV) or mass spectrometry (LC-MS, BPC) of two extracts and the corresponding locations of α-mangostin should also be marked. Also, major peaks should be identified and discussed for figure 3.
- It is suggested to briefly describe the process of mass spectrometry (LC-MS) identification of α-mangostin.
- Add references in line 228-229.
- Line 318, 1 N NaOH?
- Line 333, what proportion of DMSO is used to dissolve the sample?
- Like line 333 and 334, 20 µl change to 20 µL; 100 mg/ml change to 100 mg/mL, please review the full text and make all revisions.
- Line 379, lack of a test method for membrane diameter.
- The English editing needs to be much improved.
Author Response
Response Letter
Manuscript ID: molecules-1178441
Title: Extraction of Tropical Fruit Peels and Development of HPMC Film Containing the Extracts as an Active Antibacterial Packaging Material
Special issue: Circular Economy: Agri-Food Byproducts as Source of Bioactive Compounds
Comments of reviewer 3
This manuscript entitled “Extraction of Tropical Fruit Peels and Development of HPMC Film Containing the Extracts as an Active Antibacterial Packaging Material”. In this study, the peels of mangosteen (Garcinia mangostana), rambutan (Nephelium lappaceum), and mango (Mangifera indica) were extracted to obtain bioactive compound by microwave-assisted extraction (MAE) and maceration with water, ethanol 95% and ethanol/water. Meanwhile, the effect of antibacterial of HPMC film containing three tropical fruit peels extractions was researched on Staphylococcus aureus (S. aureus) and Escherichia coli (E. coli). The result of research shows that the mangosteen extracts have a potential to be used as an antibacterial compound in an active packaging. The work is meaningful, but some detailed comments are made as below:
Response: The authors thank the reviewer 3 for help to improve the manuscript and we try to answer for all questions.
Detailed comments:
1. Line 45-52, add references.
Response: The references were added for sentences in line 44 – 51 (please see line 44 -51).
2. There is some confusion in the abbreviation in the article. It is suggested to choose a suitable abbreviation and keep the whole text unified. Like Mi is the same with microwave-assisted extraction (MAE) from my perspective.
Response: To be clearer, MAE was used as the abbreviation for microwave-assisted extraction and Mi was changed to MAE in the manuscript (please see the manuscript).
3. Delete the sentence in line 88-92, consider its relevance to the topic.
Response: The sentence in line 88 – 92 was deleted from the revised manuscript.
4. The p value was mentioned in line 96 and line 144, but the results of statistical analysis are not shown in the table 1 and Figure 2.
Response: The statistical analysis of Data was added in Table 1, and Figure 1 and 2 (please see revised Table 1, and Figure 1 and 2).
5. Line 115-121, it is suggested to put two sentences in the introduction section. Although microwave-assisted extraction (MAE) has advantages over traditional methods, the advantages of MAE in extracting phenolic acids cannot be directly proved especially this part is the result of phenolic content.
Response: The sentences in line 115 – 121 was moved to the introduction (please see line 66 -72).
6. It is recommended that the figures are further optimized to ensure its resolution, such as Figure 1, 2 and 3.
Response: The resolution of all Figures was improved to be better.
7. Results descriptive and inferential sentences should be used with caution, such as lin177-184. Please review the full text.
Response: Results descriptive and inferential sentences have been reviewed again throughout manuscript.
8. It is recommended to add chromatography (HPLC, UV) or mass spectrometry (LC-MS, BPC) of two extracts and the corresponding locations of α-mangostin should also be marked. Also, major peaks should be identified and discussed for figure 3.
Response: The LC-MS chromatograms of MT-MAE-E/W and MT-Ma-E/W were provided in supplementary. Mass-spectra of both extracts was marked at the location of α-mangostin in red color. Other peaks were identified as well but they relate with impurity or unknown substances. Some peaks could not identified by software library. However, overall mass-spectra and some peaks were explained in the manuscript (please see line 214 -222).
9. It is suggested to briefly describe the process of mass spectrometry (LC-MS) identification of α-mangostin.
Response: The α-mangostin was identified by software library. It was confirmed by matching score which must be more than 90%, and previous studies (please see line 204 - 206 and 406 - 408).
10. Add references in line 228-229.
Response: The reference [31] was added in line 267 in the revised manuscript (please see line 267).
11. Line 318, 1 N NaOH?
Response: N is Normality which is a concentration unit. 1 N NaOH is equivalent to 1 molar (mol/L) NaOH.
12. Line 333, what proportion of DMSO is used to dissolve the sample?
Response: Due to the low solubility of the fruit extracts in ethanol, we thus used 100% DMSO as a solvent to make a working stock of fruit extracts. In this regard, the stocks were made at a concentration of 10 mg/mL, and 100 mg/mL for the analysis.
13. Like line 333 and 334, 20 µl change to 20 µL; 100 mg/ml change to 100 mg/mL, please review the full text and make all revisions.
Response: Abbreviation of liter was changed from l to L in all text in the manuscript (please see the revised manuscript).
14. Line 379, lack of a test method for membrane diameter.
Response: The detail explaining diameter measurement was added (please see line 423-425).
15. The English editing needs to be much improved.
Response: Thank you very much. With respect to the comment, we have revised our manuscript by our team and are hopeful that it is a better version of the manuscript originally submitted.
Round 2
Reviewer 1 Report
The manuscript, Extraction of Tropical Fruit Peels and Development of HPMC Film Containing the Extracts as an Active Antibacterial Packaging Material, is suitable for publication in Molecules, after the authors have addressed the following comments:
- In Table 1, some elements (letters a, b, ...) were added to the “Yield (%)” column. Please add a table note, such as the footnote to table 2.
- Item: 2.3 Xanthones and flavonoid content of the fruit peel extracts
- Line 149: “The flavonoids xanthones founded in mangosteen peel were mangostin, garcinone-E, gartanin and smeathxanthone-A, whereas mango peel contains mangiferin and quercetin derivatives as major flavonoids”.
- The structures of xanthones and quercetin could be added.
- The complete LC/MS chromatogram should be added, mainly the MT-MAE-W/E and MT-Ma-W/E.
Author Response
Comments of reviewer 1
The manuscript, Extraction of Tropical Fruit Peels and Development of HPMC Film Containing the Extracts as an Active Antibacterial Packaging Material, is suitable for publication in Molecules, after the authors have addressed the following comments:
Response: The authors thank the reviewer 1 for helping us to improve the manuscript and we try again to improve our manuscript.
Detailed comments:
1. In Table 1, some elements (letters a, b, ...) were added to the “Yield (%)” column. Please add a table note, such as the footnote to table 2.
Response: The description of a statistic analysis was added at the footnote of Table 1. (please see line 106 -107).
2. Item: 2.3 Xanthones and flavonoid content of the fruit peel extracts
Response: The topic 2.3 in the revised manuscript was changed to “Xanthones and flavonoid content of the fruit peel extracts” (please see topic 2.3 line 134).
3. Line 149: “The flavonoids xanthones founded in mangosteen peel were mangostin, garcinone-E, gartanin and smeathxanthone-A, whereas mango peel contains mangiferin and quercetin derivatives as major flavonoids”.
Response: The word “flavonoid” in line 149 was changed to “xanthones” (please see line 150).
4. The structures of xanthones and quercetin could be added.
Response: The structure of xanthones and quercetin were added as Figure 3 in the revised manuscript (please see Figure 3).
5. The complete LC/MS chromatogram should be added, mainly the MT-MAE-W/E and MT-Ma-W/E.
Response: The full LC/MS chromatograms of MT-MAE-W/E and MT-Ma-W/E (Figure S1) were added in supplementary data.
Reviewer 3 Report
Since the authors have addressed most of the comments I raised, and the revised version is much improved now. Thus, acceptance of the work is suggested.
Author Response
Comments of reviewer 3
Since the authors have addressed most of the comments I raised, and the revised version is much improved now. Thus, acceptance of the work is suggested.
Response: We are so thankful for the reviewer 3.